# Advances on Hormones in Cosmetics: Illegal Addition Status, Sample Preparation, and Detection Technology

**DOI:** 10.3390/molecules28041980

**Published:** 2023-02-20

**Authors:** Mengyue Li, Li Wang, Min Wang, Hua Zhao, Fengnian Zhao

**Affiliations:** College of Chemistry and Materials Engineering, Beijing Technology and Business University, Beijing 100048, China

**Keywords:** hormones, illegal addition, cosmetics, sample preparation, detection technology

## Abstract

Owing to the rapid development of the cosmetic industry, cosmetic safety has become the focus of consumers’ attention. However, in order to achieve the desired effects in the short term, the illegal addition of hormones in cosmetics has emerged frequently, which could induce skin problems and even skin cancer after long-term use. Therefore, it is of great significance to master the illegal addition in cosmetics and effectively detect the hormones that may exist in cosmetics. In this review, we analyze the illegally added hormone types, detection values, and cosmetic types, as well as discuss the hormone risks in cosmetics for human beings, according to the data in unqualified cosmetics in China from 2017 to 2022. Results showed that although the frequency of adding hormones in cosmetics has declined, hormones are still the main prohibited substances in illegal cosmetics, especially facial masks. Because of the complex composition and the low concentration of hormones in cosmetics, it is necessary to combine efficient sample preparation technology with instrumental analysis. In order to give the readers a comprehensive overview of hormone analytical technologies in cosmetics, we summarize the advanced sample preparation techniques and commonly used detection techniques of hormones in cosmetics in the last decade (2012–2022). We found that ultrasound-assisted extraction, solid phase extraction, and microextraction coupled with chromatographic analysis are still the most widely used analytical technologies for hormones in cosmetics. Through the investigation of market status, the summary of sample pretreatment and detection technologies, as well as the discussion of their development trends in the future, our purpose is to provide a reference for the supervision of illegal hormone residues in cosmetics.

## 1. Introduction

With rapid economic development, the enhancement of consumer awareness, and the trend of taking care of health and body, the community is starting to use more and more cosmetics. Although the cosmetic industry is facing opportunities, cosmetic safety is still facing great challenges. According to “Regulations on the Supervision and Administration of Cosmetics in China” and “Cosmetic regulations of the European Union (regulation 1223/2009)”, some substances, such as lead, mercury, arsenic and their compounds, phthalate plasticizers, antibiotics, hormones, nitrosamines, carcinogenic substances, as well as dioxane are forbidden components in cosmetics [1,2,3]. However, there are still many illegal cosmetics containing prohibited substances in recent years. In addition to the potential sources from unqualified raw materials, cases of artificial addition have frequently appeared. For example, in order to achieve the effects of whitening and acne removal in a short time, some cosmetics would illegally add heavy metals, hormones, antibiotics, and other drugs [4,5,6,7,8]. Different from ordinary drugs, the daily dose and frequency of cosmetics mainly depend on personal habits. Hence, once illegal cosmetics are used without the consumer’s knowledge, it is easy to have adverse reactions, such as skin problems or even abnormal growth, which can cause serious damage to human health. Therefore, the detection of forbidden components plays an important part in cosmetic safety.

Among forbidden components, hormones are one of the main reported forbidden components in cosmetics in recent years. Different from pesticides and other forbidden components, hormone residues are generally attributed to the artificial addition in cosmetic products, instead of the residue in cosmetic raw materials. Hormones are a class of widely used and significantly therapeutic drugs. In clinical applications, hormones have anti-inflammatory, anti-allergic, anti-viral, and anti-shock effects, which have been used in the treatment of rhinitis, asthma, and lung disease [9,10,11,12,13]. In the treatment of skin diseases, they are generally used to relieve skin problems, such as psoriasis and eczema. It is worth noting that hormones should only be used under medical supervision. Otherwise, the mindless usage of hormones would cause problems such as hormone-dependent dermatitis [14]. For this reason, the efficient determination of hormones in cosmetics is crucial for cosmetic safety.

Cosmetics are very complex matrices, containing large numbers of substances with various properties. In order to concentrate the analytes and minimize matrix effects, the previous sample pretreatment before instrumental analysis is usually required. In recent years, there have been several reviews reporting the commonly used sample preparation technologies for cosmetics, such as microextraction (ME), matrix solid-phase dispersion (MSPD), solid-phase extraction (SPE), and ultrasound-assisted extraction (UAE) [7,15,16,17,18,19,20]. However, the above analytes involve both essential ingredients and restricted/prohibited substances, and analytical methods are not discussed in detail. In addition, there are some comments on the analytical methods for special types of ingredients in cosmetics, such as oils [20], fragrances [21], parabens [22], dyes [23], and endocrine-disrupting chemicals [24]. Even though the pretreatment and analytical methods are summarized in detail, the analytical methods are mainly covering traditional confirmation techniques, such as liquid chromatography (LC) methods and LC with mass spectrometry (LC-MS) or tandem mass spectrometry (LC-MS/MS), rare attention is paid on the emerging analytical technologies represented by rapid detection approaches [2,19,25].

In order to give the readers a comprehensive overview of the hormone risk and illegal addition status in cosmetics, we investigate the non-conformity announcements of illegal cosmetics from the Chinese National Medical Products Administration (2017–2022) and analyze the status of illegal addition of hormones in cosmetics, such as hormone types, proportions, and detection values, as well as illegal cosmetic types in this review. Although people have realized the importance of hormone detection in cosmetics, there is a lack of detailed introduction and discussion of hormone analysis methods in cosmetics. In this respect, we summarize the application of advanced technologies such as UAE, SPE, ME, MSPD, and cloud point extraction (CPE) in recent ten years (2012–2022), as well as the application of confirmation techniques and rapid detection technologies such as enzyme-linked immunosorbent assay (ELISA) and lateral flow immunoassay (LFIA) in cosmetic hormone analysis (Figure 1). In addition, the development trends in the future, such as the exploration of novel sample preparation adsorbents and new analytical methods for hormone analysis in cosmetics, are also discussed. This paper aims to give readers a clear understanding of hormone analysis in cosmetics and provide technical support for the supervision and detection of hormone residues that may exist in cosmetics.

## 2. Hormone Risk and Market Status in Cosmetics

### 2.1. Hormones and Their Risks in Cosmetics

As biologically active substances, hormones are synthesized by highly differentiated endocrine cells and secreted directly into the bloodstream to transmit chemical messages, which in turn influence the physiological activities of the body by regulating the metabolic activities of various tissue cells [26]. Among various hormones, steroid hormones are mainly applied to the skin. According to the pharmacological effects, steroid hormones can be divided into adrenal cortical hormones and sex hormones. Furthermore, adrenal cortical hormones can be further divided into glucocorticoids and salt-metabolizing corticotropic hormones [27]. The above hormones can only be used following the doctor’s advice, otherwise, abuse or misuse would cause problems, such as dependence dermatitis. However, in order to give skincare products visible effects and thus reap huge benefits, the illegal addition of hormones in cosmetics has happened in recent years [28,29]. Therefore, hormones, represented by glucocorticoids and sex hormones, are the main focus of cosmetic supervision.

Glucocorticoids play important roles in maintaining homeostasis and normal organ function in the body. It has been reported that glucocorticoids are widely used in endocrine, respiratory, hematological, and rheumatic immune diseases, as well as in the treatment of skin diseases, such as eczema, dermatitis, and psoriasis [9,30,31]. However, the use of glucocorticoids may be accompanied by adverse reactions, as well as possible reactions or rebounds after drug withdrawal. To pursue fast action and profit, some vendors illegally add glucocorticoids into cosmetics. Even though they can achieve anti-inflammatory and anti-allergic effects in the short term, long-term usage would lead to thinning skin, dry and peeling skin, skin inflammation, and even induce skin cancer [28,30]. Sex hormones have the function of controlling sexual organs and secondary sexual characteristics, and are also involved in the basic metabolic activities of living substances, such as sugars, fats, proteins, and inorganic salts in the body, which have been widely used for the treatment of infertility and gynecologic diseases, such as functional disorders and uterine bleeding [32,33]. Even though some cosmetics illegally adding sex hormones could produce short-term benefits, such as skin whitening, wrinkle reduction, and hair growth promotion, long-term usage would cause skin problems, such as thinning and atrophy of the skin, and even increase the risk of breast cancer and hysteromyoma for women [34]. The typical hormones and their risks in cosmetics are listed in Table 1. As mentioned previously, the habits and dosages of cosmetics are difficult to control for each customer. Once cosmetics containing hormones are used excessively, consumers are prone to experience adverse reactions. Therefore, it is important to detect hormones in cosmetics for the supervision of the cosmetics market and the protection of the personal safety of consumers.

### 2.2. Hormone Addition Status in Cosmetics

Driven by interests, some traders would illegally add hormones into cosmetics. Without consumers’ knowledge, hormones would be misused and abused, which could easily cause skin problems, and ultimately threaten human health. For this reason, the national regulatory authority conducts routine inspections of potentially prohibited components in cosmetics by means of random inspections and unannounced inspections every year. China has already been the second-largest cosmetics consumer market. Therefore, the survey of Chinese cosmetic status has instructive significance for cosmetic regulation in the world. From the non-conformity announcements of cosmetics on the website of the Chinese National Medical Products Administration in the recent six years (2017–2022), it is clear that the substandard cosmetic batches have a significant downward trend with the intensification of national supervision. However, there are still 63 batches of substandard cosmetics containing hormones (Figure 2a), and two kinds of hormones are detected in 10 batches of substandard cosmetics. Based on the statistical data, the detection values of hormones are in the range of 0.1 to 1385.2 μg g^–1^ (Figure 2b). Among them, the detection value of betamethasone 17-valerate is the highest, which is even over the content in skin clinical drugs. In addition, clobetasol 17-propionate, betamethasone and its derivatives (i.e., betamethasone 17-valerate, betamethasone 21-valerate, and betamethasone dipropionate), as well as triamcinolone acetonide and its derivatives (i.e., triamcinolone acetonide acetate) were the most frequently found hormones in cosmetics, whose proportions are 39.7%, 17.8%, and 13.7%, respectively (Figure 2c). It is worth noting that, compared with other cosmetic products, the facial mask is a potentially high-risk cosmetic (Figure 2d). Therefore, it is necessary to increase the monitoring of facial masks in the future.

## 3. Cosmetic Sample Preparation Technologies

Cosmetics are extremely complex and may contain lipophilic or polar, basic, acidic or neutral components. Otherwise, there are various cosmetic dosage forms, such as solid, colloid, and emulsified conditions. Due to the complexity of cosmetic ingredients and the low concentration of hormones, it is necessary to carry out efficient pretreatment before instrumental analysis, which plays an important role in the separation and collection of hormones from the complex cosmetic matrix. In this section, we will introduce some advanced technologies and discuss their development trends in cosmetic pretreatment, including UAE, SPE, ME, and other pretreatment technologies.

### 3.1. UAE

UAE is a classic extraction technology, which uses the mechanical, cavitation and thermal effects of ultrasound to extract the active compounds in the matrix [35,36]. This technique can shorten extraction time and save solvent amount, which has been applied in food and natural products [37,38]. The factors affecting the extraction efficiency of the UAE are ultrasound frequency, intensity, time, and temperature. According to the literature, acetonitrile (ACN), methanol (MeOH), and tetrahydrofuran (THF) can be used as the extraction solution for ultrasonic treatment for 20~30 min for hormones in cosmetics [39,40,41]. Compared with ACN, MeOH can cause cosmetic emulsification easily, therefore ACN is more widely used to simplify the procedure and maintain its accuracy [9]. To further reduce the emulsification of cosmetic products, saturated sodium chloride (NaCl) solution can also be added during the extraction process. After the extraction process, centrifugation and filtration steps are commonly necessary.

Based on UAE, Xu et al. [42] extracted hydrocortisone, prednisolone, prednisone, dexamethasone, hydrocortisone acetate, cortisone acetate, prednisolone acetate, and triamcinolone acetonide from body lotion and creams by sonication in MeOH for 20 min at 20 °C. Wu et al. [41] added NaCl solution and ACN in lotions and creams, followed by sonication at 25 °C for 30 min. Coupled with LC–MS/MS, the limit detections (LODs) of hydrocortisone, estrone, canrenone, triamcinolone acetonide, and progesterone were less than 25 pg and the recoveries ranged from 90.6% to 118.0%. Fiori et al. [39] adopted THF as the solvent to extract betamethasone 17-valerate, beclomethasone, beclomethasone dipropionate, and methylprednisolone in cosmetics by sonication for 20 min. Coupled with LC–MS, the LODs for the above six glucocorticoids were from 12.1 to 35.4 mg L^–1^ and the recoveries were from 92% to 98% in cream samples.

### 3.2. SPE

SPE is an advanced sample pretreatment technique in modern analytical chemistry, in which the targets present in the sample are generally retained on the solid phase extraction material by adsorption, ion exchange, ligand, or other chemical interactions, thus achieving the separation of the analytes from the matrix [43,44]. There are two types of SPE technique. One is to retain the analytes on solid sorbents and the other is the interference. Due to the high selectivity and potential recyclability, the first one is the most widely used type. As shown in Figure 3a, the main operation steps may include conditioning (or activation), loading, washing, and elution [15]. This method is simple, rapid, and environmentally friendly, and has been applied in environmental monitoring, food, and drug analysis [45,46,47].

Currently, the most commonly used adsorbents in SPE are normal-phase materials (Florisil, alumina, silica, and sea sand) and reverse-phase materials (C_18_ and C_8_ silica) [15]. In order to improve the affinity with analytes, researchers proposed selective sorbents, such as molecular imprinting polymers (MIPs) [48]. These polymers are porous materials with selective binding cavities for the specific recognition of a particular analyte or class of chemically related analytes, which can also be called “artificial antibodies” [49,50,51]. For instance, Wang et al. [52] developed a MIP-based SPE material for prednisone capture in cosmetics. The polyethylene filter plate was coated with multi-walled carbon nanotubes (MWCNTs), and then MIPs were prepared based on the surface imprinting technique (Figure 3b). These materials (plate@MWCNTs@MIPs) can simplify the experimental steps, save time and costs, and can be recycled. Coupled with high-performance LC (HPLC), this approach could be used for the selective prednisone separation, purification, and detection in mask, moisturizer, masque, and milk samples with satisfactory recovery (83.0–106.0%) and low LOD (5.0 μg L^–1^).

Even though it is high efficiency, the sorbent bets of SPE (cartridges, pre-columns, or disks) would cause high mass transfer resistance, which may cost a long time and a large amount of solvents during the elution process. For this reason, the dispersive SPE (dSPE) is developed as a supplement [3,53,54]. Based on magnetic or magnetizable sorbents, magnetic solid-phase extraction (MSPE) has also been applied, which makes the recovery under a magnetic field easier and faster [55,56]. For instance, Zhao et al. [57] proposed an MSPE method using magnetic MWCNTs (MMCNTs) as the sorbents. Followed by HPLC, this method could be used for the rapid and efficient extraction of four sex hormones in toners, with recoveries ranging from 80.1% to 118.8%. Magnetic metal-organic frameworks-101 functionalized with graphite-like carbon nitride materials (Fe_3_O_4_/g-C_3_N_4_/MIL-101) were also fabricated [58]. These materials show excellent selectivity for glucocorticoids due to the hydrogen bonding effect with g-C_3_N_4_ and the size-matching effect with MIL-101, which could be used as MSPE extractants for glucocorticoids. Coupled with ultra-performance LC-MS/MS (UPLC-MS/MS), this method could be applied to the determination of glucocorticoids in facial masks and toners. In order to further improve the selectivity further, researchers also proposed magnetic MIPs (MMIPs, Fe_3_O_4_@SiO_2_-MIP), which used dexamethasone and hydrocortisone as the templates [59,60]. As shown in Figure 3c, the MMIPs showed a higher adsorption amount for the template molecule than its structure analog. Combined with HPLC, this method could realize the rapid and selective extraction and determination of dexamethasone and hydrocortisone in skincare cosmetic samples.

### 3.3. ME

ME allows the extraction and concentration of analytes in extractants with a volume of less than 100 μL [61,62]. Hence, this technique is more eco-friendly and has lower LODs than traditional extraction technologies, which has been applied for multiple-compound extraction (e.g., volatile and non-volatile, polar and nonpolar, ionic or metallic species) in the complex matrix [24]. Based on the phases, microextraction techniques can be classed into liquid-phase ME (LPME) and solid-phase ME (SPME) [17,63,64].

Due to their designed physical and less-harmful properties, some friendly alternatives, such as ionic liquids (ILs), have been explored as extractants for LPME [64]. As shown in Figure 4a, the IL homogeneous ME (LLME) method was performed using hydrophilic 1-hexyl-3-methylimidazolium tetrafluoroborate ([C_6_MIM][BF_4_]) as extraction solvent and ammonium hexafluorophosphate (NH_4_PF_6_) as the ion-pairing agent [65]. Coupled with HPLC, this method can be applied for eight hormones detection in liquid and gel-like cosmetics, such as 17α-estradiol and estrone, with the low LODs (0.03–0.24 μg L^–1^) and good recoveries (96.3% to 111.4%). As the new subclass of ILs, magnetic ILs (MILs), such as [P_6,6,6,14_^+^]_2_[CoCl_4_^2−^], were also synthesized by the incorporation of a paramagnetic component [66]. Adopting the dispersive LLME (DLLME) coupled with HPLC analysis, this method can be used for the extraction and detection of six estrogens in lotions, with the LODs not over 15 ng mL^−1^ and recoveries ranging from 96.3% to 111.4% (Figure 4b).

Using porous monolithic polymer as an extraction medium, polymer monolith ME (PMME) technology integrates extraction, purification, and enrichment, which has the advantages of continuous porosity, wide pH application range, high extraction capacity, and good stability. Generally, PMME columns are prepared by in situ polymerization of a mixture of functional monomers (e.g., methacrylic acid-co-ethylene glycol dimethacrylate, butyl methacrylate-co-ethylene dimethacrylate), crosslinking agents and pore-causing agents. Combined with HPLC, some modified PMME columns have been used as the extraction media for the determination of sex hormones, including estrogen, testosterone, methyltestosterone, and progesterone in cosmetics with the low LODs (not over 4.6 μg L^–1^) [67,68]. Based on the above features, Wei et al. [69] prepared a porous monolithic polymer inside fiber by in situ photopolymerization combined with sacrificial support in hollow fiber (Figure 4c). Coupled with HPLC, this fabricated micro-SPE (µ-SPE) device could be used for prednisone acetate, prednisone and prednisolone determination in lotions with the LOD of 1.5 μg L^–1^ and recoveries of 69.0–113.3%.

### 3.4. Other Pretreatment Technologies

Since first reported in 1989, MSPD has been widely used for the one-step extraction and purification of organic analytes in various fields [63,70]. Different from SPE or SPME, the samples and sorbents are mixed and grinded directly. Without the dissolving or dispersing of samples into solvents in advance, MSPD integrates extraction and clean-up processes in one step, which can eliminate matrix interference and reduce solvent consumption [16,71]. The commonly used solid support materials are the same as those in SPE [15]. For example, Guo et al. [72] adopted MIPs as the MSPD adsorbents for the special and selective extraction of dexamethasone and hydrocortisone in cosmetics. Coupled with HPLC, this method can be validated for dexamethasone and hydrocortisone analysis in cosmetics samples with good selectivity, sensitivity, and efficiency, in which the LODs are of 0.03 μg g^–1^ for dexamethasone and 0.02 μg g^–1^ for hydrocortisone, respectively.

Compared with the traditional liquid–liquid extraction, CPE is a more eco-friendly tool, which can be used to extract/preconcentrate and analyze hydrophobic analytes, or can be converted to hydrophobic analytes, because nonionic surfactants will form micelles in aqueous media when heated above cloud point temperature or added with salt (salting-out phenomenon) [73]. Currently, non-ionic surfactants (e.g., Triton X-114 and Triton X-100), ILs, polyethylene glycols, and anionic surfactants have been applied for CPE [15]. To shorten the equilibration time of estrogens in the CPE procedure, Xiao et al. [74] developed a rapid and efficient co-precipitation-assisted CPE (CpCPE) technique based on the combination of co-precipitation with aluminum hydroxide and CPE with sodium dodecyl sulfate. Followed by HPLC analysis, this method can be achieved for five estrogens determination in toner samples, including 17β-estradiol, estrone, ethinyl estradiol, diethyl stilbestrol, and dihydro stilbestrol. The recoveries ranged from 77.3–104.1% with RSD of 2.0–10.4%, and the LODs were not over 0.7 μg L^–1^. This method was suitable for the rapid and sensitive analysis of trace estrogens.

## 4. Analytical Technologies for Hormones in Cosmetics

The determination of the hormones in cosmetics is usually carried out by confirmation techniques, such as LC and LC-MS or LC-MS/MS. In order to satisfy the demand for rapid and on-site determination, some rapid detection methods have come out. In this section, we discuss the widely used analytical technologies for hormones in cosmetics. Detailed information including pretreatment and analytical technologies for hormones in cosmetics is listed in Table 2 and Table 3.

### 4.1. Confirmation Technologies

LC is a chromatographic system, in which the liquid is used as the mobile phase, and the pressure is increased by the high-pressure infusion pump to improve efficiency. Common detectors used in LC include UV-VIS detectors and diode array detectors (DAD), which use the principle that a physical or chemical property of the sample differs from the mobile phase to achieve the separation of substances [75,76]. This technique has been used to detect polar compounds with high sensitivity, selectivity, and reproducibility [77,78]. The choice of mobile phase and chromatographic column are the main two important parts. Generally, the mobile phase should have a different polarity from the chromatographic column. Due to the polar properties, the reverse phase LC system is suitable for hormone separation, including a non-polar chromatographic column and polar mobile phases. According to the literature, C_18_ columns are the widely used reversed-phase columns for hormone separation. In order to enhance the separation efficiency for special targets, the multi-dimensional liquid chromatographic system has been used for hormone detection. For example, a two-dimensional liquid chromatography method consisting of a molecular-imprinted monolithic column coupled with a C_18_ column (MIMC-2D-LC) was also developed for estradiol analysis in cosmetic samples [34]. Generally, ACN/H_2_O or MeOH/H_2_O are generally performed as the mobile phases. To help the subsequent ionization of the analytes and consequently enhance the detection signals, formic acid (FA), acetic acid (Ac), ammonium formate (NH_4_OFA), ammonium acetate (NH_4_OAc), phosphate, or ammonium bicarbonate (NH_4_HCO_3_) at low concentrations would be added into the aqueous phase or organic phase.

MS can separate the various components into ions with different charge-to-mass ratios, generate an ion beam by accelerating the electric field, and use a mass analyzer to detect data such as chemical structure and molecular mass [76,79,80]. LC-MS or LC-MS/MS makes full use of the high qualitative performance of MS and the high separation performance of LC, allowing the simultaneous identification and detection of multiple target analytes at trace levels. Generally, the choice of mobile phase and chromatographic column for LC-MS or LC-MS/MS follows the same principle as LC. Based on UPLC-MS/MS, Meng et al. [53] developed a broad screening method for 100 illicit ingredients (including 40 glucocorticoids and 8 sex hormones) in cosmetics using the C_18_ column as the analytical column. Two binary mobile phase compositions were used for analyses run in positive and negative electrospray ionization (ESI^+^ and ESI^−^) modes (ACN/5 mM NH_4_OFA solution for the ESI^+^ mode, and ACN/5 mM NH_4_HCO_3_ solution for the ESI^−^ mode). After employing UAE and dSPE as the preparation procedures, this method can realize the identification and quantitation of 48 hormones in lotions and creams, with the LODs of 1.1–12.5 μg kg^–1^.

### 4.2. Rapid Detection Methods

Although the above methods have high sensitivity and accuracy, they are high cost, require professional operators and complicated pretreatment technologies, and cannot meet the demand for on-site and rapid detection of hormones in cosmetics. In this regard, many efforts have been carried out to explore rapid and reliable immunoassays, such as ELISA [88,89] and LFIA technologies [90], which could act as important supplements for the current analytical means.

Based on antigen-antibody immunoreactions and enzyme catalysis, ELISA is the gold standard technology, which has been used in clinical diagnosis, food quality control, and environmental monitoring [91,92]. As a rapid and inexpensive tool for compound residues, ELISA also shows satisfying sensitivity and only needs easy pretreatment procedures. To realize the supervision and control of the hormone additives in large series of cosmetic samples, Zhang et al. [86] established a heterologous competitive indirect ELISA method for hydrocortisone detection. Herein, the polyclonal antibodies were obtained by immunization after the conjugation of hydrocortisone with bovine serum albumin, and progesterone carboxylic acid acted as a competitive hapten. Only adopting the centrifugation, ultrasound-assisted extraction, and five-fold working buffer dilution steps, this assay could be used for the determination of hydrocortisone in lotion, cream, and toner samples with the LOD of 0.04 ng mL^−1^ and recoveries of 82.5% to 118.9%. Compared with the standard LC–MS/MS, this method appeared good correlation, indicating its potential application in hydrocortisone monitoring in cosmetics.

LFIA is a very simple, rapid, and portable analytical device that combines immunoassay technology with chromatography. Typically, this device is composed of a sample pad, conjugated pad, nitrocellulose membrane, and absorption pad [93,94]. As a paper-based chromatography, LFIA can transport a fluid sample across various strip zones under the capillary force, which can be used for the immediate detection of antibodies or antigens via antibody-antigen interactions [95]. The conventional immunoassay format approaches and the LFIA principle for on-site applications are shown in Figure 5a. Based on the rapid and flexible features of LFIA, our group [85] developed a bare eye-based semiquantitative colloidal gold test strip for the rapid detection of dexamethasone in commercial facial masks (Figure 5b). According to the colloidal gold aggregation precipitation chromogenic reaction, this LFIA-based test strip could be directly viewed by eyes towards dexamethasone with the concentration of 10 to 200 ng mL^–1^. To enhance the sensitivity and accuracy, a fluorescent reader-based test strip was developed for the quantitative detection of triamcinolone acetonide in cosmetics using up-conversion nanoparticles (UCNPs) as the probe [87]. Using dexamethasone derivative as a coating antigen, this assay provides an easy (only diluting samples with water), fast (10 min including pretreatment), wide linear range (1.0–100 ng mL^−1^), low LOD (20 μg kg^−1^) and low-cost method, which can be applied for the on-site determination of triamcinolone acetonide in cream, mask and essence samples (Figure 5c). In order to further achieve the accurate quantitative analysis for glucocorticoids, Zhang et al. [86] also proposed the bifunctional LFIA test strip for the direct visual or quantitative detection of dexamethasone based on UCNPs. This strategy exhibits a favorable detection range of 0.1–9 ng mL^−1^, with the LOD of 2.0 μg kg^−1^ for dexamethasone in food supplements and cosmetic samples. Using the optical density scanner, more accurate quantitative data could be obtained, which offers a new avenue for the visual quantitative analysis of glucocorticoids in cosmetics.

## 5. Conclusions and Outlooks

As a sunrise industry, the cosmetic industry has not only ushered in the opportunity for rapid development, but also faces new challenges, especially cosmetic safety. According to the data from China in 2017–2022, hormones are still the main prohibited substances in illegal cosmetics. Even though the frequency of hormone additions in cosmetics has declined significantly, the community still needs to pay close attention to potentially banned substances in cosmetics due to the growing expansion of sales channels represented by e-commerce. Currently, many efforts have been carried out to develop reliable analytical methods for the determination of hormones in cosmetics. Due to the complex matrices, efficient sample pretreatment is necessary before analysis. At present, UAE, SPE, and ME are still widely used pretreatment techniques. The current analytical methods for hormones in cosmetics are still limited to standard methods, such as LC or LC-coupled MS methods. Only several kinds of literature report immunological-based rapid methods. Therefore, there is still a lack of reliable and rapid analytical methods for hormones in cosmetics.

In the future, researchers will explore more environmentally friendly extraction approaches, combining miniaturization and portable extraction equipment to reduce the consumption of samples, solvents, and time. In this case, with the development of nanotechnology, novel and functional nano-sorbents, such as MOFs, MIPs, and magnetic nano-sorbents will play important parts owing to their high specific surface areas and special binding sites for the efficient or selective extraction of hormones. It should be noticed that some rapid and on-site analytical strategies such as surface-enhanced Raman spectroscopy (SERS) [96,97] and electrochemical methods [98,99,100,101] have been applied for the detection of hormones in the field of food and environment. Moreover, new recognition mechanisms based on the specific binding between hormones and aptamer or other molecules are also put into application [99,102,103]. In fact, the kinds of hormones in cosmetics are similar to those in food and environment, so these strategies will provide guidance for the development of rapid detection methods for hormones in cosmetics. Finally, the whole society should make concerted efforts to ensure the quality and safety of cosmetics.

## Figures and Tables

**Figure 1 molecules-28-01980-f001:**
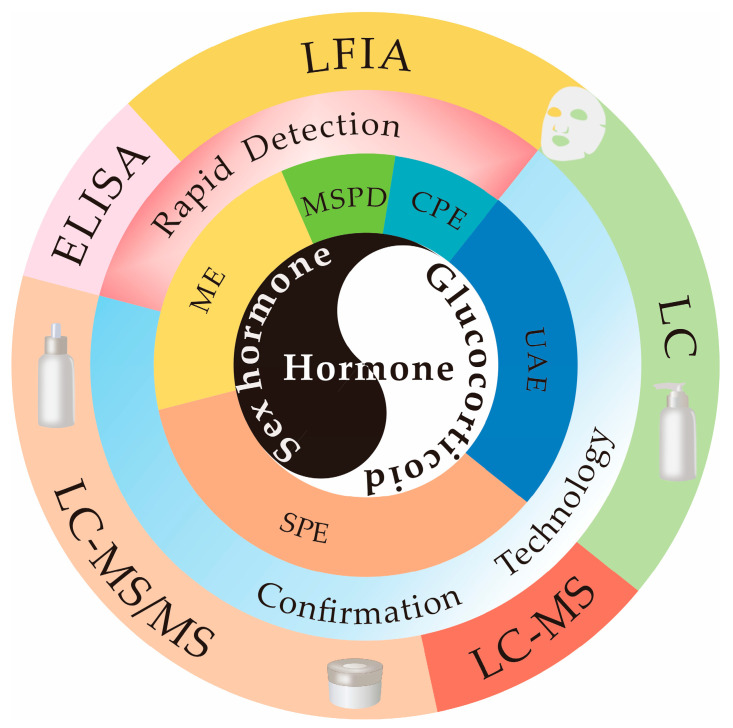
Sample preparation and detection techniques for hormones in cosmetics are included in this review.

**Figure 2 molecules-28-01980-f002:**
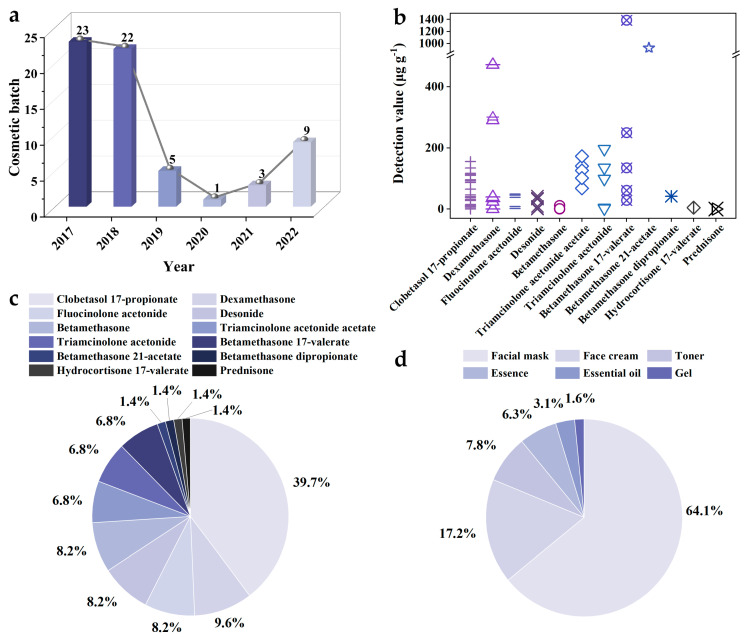
(**a**) The unqualified cosmetic batches containing hormones and (**b**) their detection values in China in recent six years (2017–2022). (**c**). Hormone types, frequency, and (**d**) cosmetic types in the above unqualified cosmetic batches.

**Figure 3 molecules-28-01980-f003:**
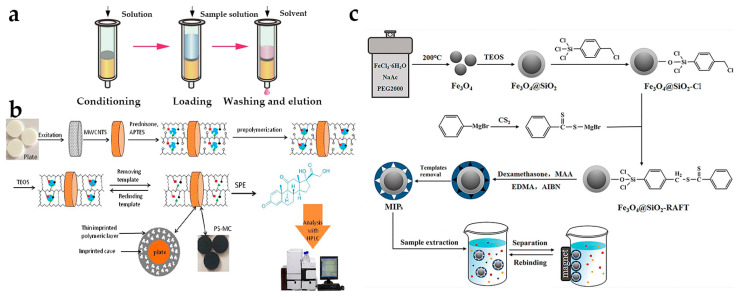
(**a**) Scheme of sorption-based extraction technique for SPE. Reprinted with permission from Ref. [15] Copyright 2017 Journal of Separation Science. (**b**) Synthesis of plate@MWCNTs@MIPs and their application for SPE. Reprinted with permission from Ref. [52] Copyright 2018 Journal of Colloid and Interface Science. (**c**) Synthesis of magnetic MIPs and their application for MSPE. Reprinted with permission from Ref. [59] Copyright 2018 Journal of Separation Science.

**Figure 4 molecules-28-01980-f004:**
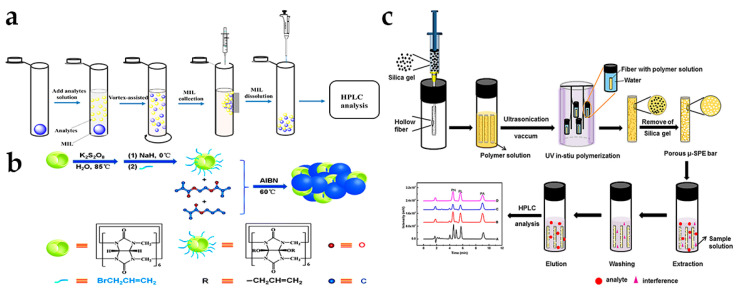
(**a**) Scheme of the MIL as an extraction solvent for DLLME. Reprinted with permission from Ref. [66] Copyright 2020 Talanta. (**b**) Scheme for the synthesis of ACB[6]@poly(BMA-EDMA) monolith for PMME. Reprinted with permission from Ref. [68] Copyright 2015 New Journal of Chemistry. (**c**) Scheme of the preparation of porous monolithic polymer extraction bars and the procedure of µ-SPE. Reprinted with permission from Ref. [69] Copyright 2020 Journal of Separation Science.

**Figure 5 molecules-28-01980-f005:**
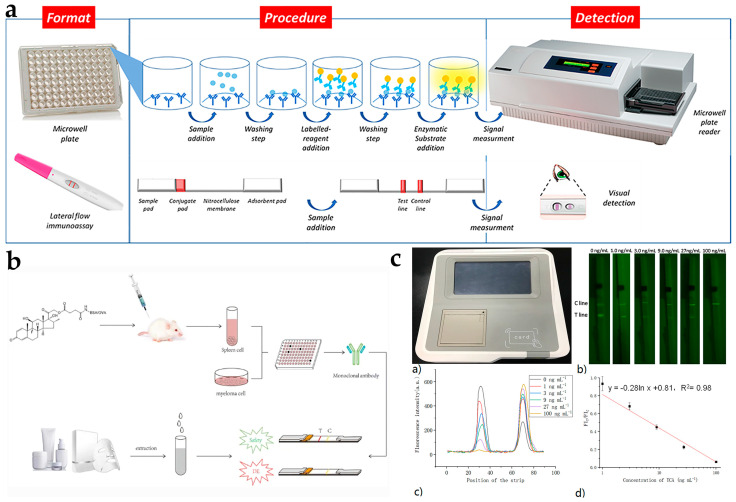
(**a**) Scheme of the conventional immunoassay format approaches and the LFIA principle for on-site applications. Reprinted with permission from Ref. [25] Copyright 2021 Processes. (**b**) The application of a LFIA assay to rapidly test for dexamethasone in commercial facial masks. Reprinted with permission from Ref. [85] Copyright 2019 Analytical and Bioanalytical Chemistry. (**c**) The UCNPs-LFIA for triamcinolone acetonide detection. Reprinted with permission from Ref. [87] Copyright 2019 Spectrochimica Acta Part A: Molecular and Biomolecular Spectroscopy.

**Table 1 molecules-28-01980-t001:** Typical hormones in cosmetics and their potential risks for humans.

Hormones	Typical Chemical Structures	Risks
Glucocorticoids	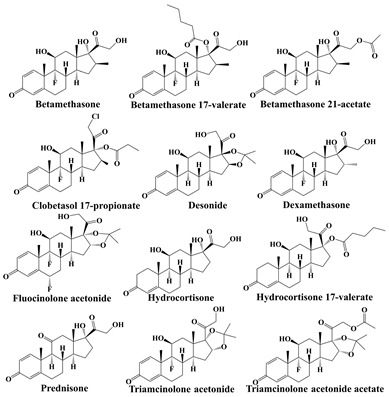	skin thinning, inflammation, dryness and peeling, and even skin cancer [28,29,30]
Sex hormones	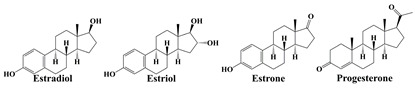	skin thinning and atrophy, even hysteromyoma and breast cancer [34]

**Table 2 molecules-28-01980-t002:** Confirmation technologies for hormone determination in cosmetics.

Hormones	Pretreatment Method	Condition/Material	Analytical Method	Mobile Phase	Stationary Phase	Linear Range μg L^–1^	LOD μg L^–1^	Sample	Ref.
Methylprednisolone beclomethasone flunisolide budesonide betamethasone 17-valerate beclomethasone dipropionate	UAE	UAE in THF for 20 min	LC–MS	0.1% FA in ACN/0.1% FA in H_2_O	C_18_	300–10000	24.2 35.4 30.1 18.5 12.1 12.4	Creams	[39]
Triamcinolone prednisolone methylprednisolone betamethasone dexamethasone fluocinolone acetonide prednisolone-21acetate hydrocortisone-21-acetate clobetasol propionate betamethasone dipropionate fluocinolone acetonide-21-acetate	UAE	UAE in ACN for 15 min at 50 °C	LC-MS/MS	ACN/0.1% FA in H_2_O	Atlantis T3	10–1000	0.25 1.0 0.3 0.3 0.25 1.0 0.25 1.0 0.25 1.0 1.0	Creams	[40]
Hydrocortisone estrone canrenone triamcinolone acetonide progesterone	UAE	UAE in ACN for 15 min at 25 °C	LC–MS/MS	0.2% FA in MeOH/0.2% FA in H_2_O	C_18_	10–1500	<25 pg	Lotions, creams	[41]
Glucocorticoids (43)	UAE	UAE in MeOH for 30 min	UPLC-MS/MS	0.1% FA in ACN/0.1% FA in H_2_O	C_18_	100–2000	0.33–30	Solutions, lotions, creams, gels, powders	[81]
Glucocorticoids (47)	SPE	Oasis HLB SPE cartridge	UPLC–MS/MS	ACN/0.1% FA in H_2_O	C_18_	–	0.05–0.4	Cream, gel, lotion, solution, powder, mask	[82]
Prednisone	SPE	plate@MWCNTs@MIPs	HPLC–UV	ACN/H_2_O	C_18_	1000–6000	5.0	Mask, masque, milk, moisturizer	[52]
Glucocorticoids (12) Sex hormones (32)	dSPE	C_18_ and MgSO_4_	UPLC–MS/MS	MeOH/1.5 mM NH_4_OAc in H_2_O	HSS-T_3_	0–480	–	Cream, emulsion, shampoo	[3]
Glucocorticoids (40) Sex hormones (8)	UAE-dSPE	C_18_ and PSA	UPLC-MS/MS	ACN/NH_4_HCO_3_ solution	C_18_	3.6–2000	1.1–12.5	Lotions, creams	[53]
Ethinylestradiol norgestrel megestrol acetate medroxyprogesterone acetate	MSPE	MMWCNTs	HPLC–DAD	ACN/0.1% phosphate aqueous solution	C_18_	200–20,000	330 60 80 80	Toners	[57]
Hydrocortisone dexamethasone desonide fluocinonide flunisolide	MSPE	Fe_3_O_4_/g-C_3_N_4_/MIL-101	UPLC–MS/MS	ACN/0.2% FA in H_2_O	C_18_	0.02–2.0 0.02–2.0 0.02–2.0 0.02–2.0 0.01–1.0	0.005 0.005 0.005 0.005 0.002	Facial masks, toners	[58]
Dexamethasone	MSPE	Fe_3_O_4_@SiO_2_-MIPs	HPLC–UV	ACN/H_2_O	C_18_	500–50,000	50	Cosmetics	[59]
Dexamethasone hydrocortisone	MSPE	Fe_3_O_4_@SiO_2_-MIPs	HPLC–UV	ACN/H_2_O	C_18_	500–15,000	15	Lotions, toners, masks	[60]
17α-Estradiol 17α-ethinylestradiol estrone 17α-hydroxyprogesterone medroxyprogesterone megestrol-17-acetate norethisterone acetate progesterone	HILME	[C_6_MIM][BF_4_]	HPLC–DAD	ACN/H_2_O	C_18_	0.625–125 0.625–125 0.625–125 0.25–50 0.25–50 0.125–100 0.25–125 0.125–125	0.24 0.19 0.18 0.08 0.09 0.03 0.05 0.03	Liquid and gel-like cosmetics	[65]
Estrone estradiol 17-α-hydroxyprogesterone chloromadinone 17-acetate megestrol 17-acetate medroxyprogesterone 17-acetate	DLLME	MILs [P_6,6,6,14_^+^]_2_[CoCl_4_^−^]	HPLC–UV	ACN/H_2_O	C_18_	40–1000 40–1000 20–1000 30–1000 30–1000 40–1000	15 15 5.0 8.0 8.0 15	Lotions	[66]
Estrone estradiol estriol progesterone diethylstilbestrol	PMME	ACB[6]@poly(BMA-EDMA)	HPLC–DAD	ACN/H_2_O	Shimpack PREP-ODS (H) and C_18_	1.0–1000 1.0–1000 1.0–1000 1.0–1000 1.0–700	0.012 0.022 0.053 0.017 0.019	Cosmetics	[68]
Dexamethasone betamethasone prednisolone triamcinolone triamcinolone acetonide cortisone hydrocortisone fluocinonide beclomethasone dipropionate	PMME	P(BMA-EDMA-GN)	LC–MS	0.3% FA in ACN/0.3% FA in H_2_O	C_18_	1.0–800	0.28 0.32 1.40 1.93 0.22 0.89 1.52 0.18 0.13	Cosmetics	[83]
Prednisone prednisolone prednisolone acetate	μ-SPE	hollow fibers	HPLC–UV	MeOH/H_2_O	C_18_	5.0–2000	1.5	Lotions	[69]
Dexamethasone hydrocortisone	MSPD	MIPs	HPLC–UV	ACN/H_2_O	C_18_	50–50,000	30 20	Cosmetics	[72]
17β-estradiol ethinyl estradiol estrone diethyl stilbestrol dihydro stilbestrol	CPE	AlCl_3_ and SDS	HPLC–UV	ACN/H_2_O	C_18_	2.0–50	0.2 0.7 0.4 0.4 0.2	Toners	[74]

UAE: ultrasound-assisted extraction; THF: tetrahydrofuran; ACN: acetonitrile; FA: formic acid; MeOH: methanol; SPE: solid phase extraction; MWCNTs: multiwalled carbon nanotubes; MIPs: molecular imprinted polymers; UV: UV-vis detector; dSPE: dispersive solid-phase extraction; UPLC: ultra-high-performance liquid chromatography; MSPE: magnetic solid-phase extraction; MMWCNTs: magnetic multiwalled carbon nanotubes; DAD: diode array detector; Fe_3_O_4_/g-C_3_N_4_/MIL-101: magnetic metal-organic frameworks-101 functionalized with graphite-like carbon nitride material; μ-SPE: micro-solid phase extraction; HILME: homogeneous ionic liquid microextraction; [C_6_MIM][BF_4_]: 1-hexyl-3-methylimidazolium tetrafluoroborate; DLLME: dispersive liquid-liquid microextraction; MILs: magnetic ionic liquids; PMME: polymer monolith microextraction; ACB[6]: allyloxy-cucurbit [6]; P(MMA-co-MAA): poly(methyl methacrylate-co-methacrylic acid); EDMA: ethylene dimethacrylate; GN: graphene; MSPD: matrix solid-phase dispersion; CPE: cloud-point extraction; SDS: sodium dodecylsulfate.

**Table 3 molecules-28-01980-t003:** Rapid analytical technologies for hormone determination in cosmetics.

Hormones	Pretreatment Method	Condition	Detection Method	Nanomaterial	Recognition Element	Linear Range μg L^–1^	LOD μg L^–1^	Sample	Ref.
Hydrocortisone	UAE	UAE for 10 min	ELISA	–	Antibody	0.1–2.0	0.04	Body lotions, moisture creams, toners	[84]
Dexamethasone	UAE	UAE in ACN and saturated NaCl solution for 10 min	LFIA	colloidal gold	Antibody	10–200	10	Facial masks	[85]
Dexamethasone	Dilution	20-fold dilution by H_2_O	LFIA	UCNPs	Antibody	0.1–9.0	2.0	Cosmetics with skin whitening and acne treatment functions	[86]
Triamcinolone acetonide	Dilution	20-fold dilution by H_2_O	LFIA	UCNPs	Antibody	1.0–100	20	Creams, masks, essences	[87]

ELISA: enzyme-linked immunosorbent assay; LFIA: lateral flow immunoassay; UCNPs: up-conversion nanoparticles.

## Data Availability

Not applicable.

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
