# Peer review of "Advances on Hormones in Cosmetics: Illegal Addition Status, Sample Preparation, and Detection Technology"

_molecules, 2023, doi:10.3390/molecules28041980_

Round 1
Reviewer 1 Report
This paper is a review of a seemingly subordinate, but in fact very dangerous aspect of cosmetics industry: the addition of hormones to cosmetics products. These additives could cause some short-time apparent advantages in the applicative goals of these products, but their long-term health effects are (could be) sources of serious health riscs, including skin cancer and similar problems. The hormone additives in cosmetics represent a difficult analytical callenge: it is the detection/determination of small quantities in a fairly complicated mixture of (mainly organic) components.
The present review gives a clear description of the background and of the major analytical techniques which can be used for the detection and/or determination of these additive, which often became applied illegally. The analysis requires two steps: sample preparation (separation) and the following determination of the dangerous substance. In the present review also the most modern techniqes are described, as ultrasound-assisted extraction, solid-phase extraction or microextraction, together with frontier determination techniques, as immunoassay.
The review is based on 100 well selected recent references, all are after 2000, mostly from the last decade.
Summarizing the opinion of the present Referee: this is an excellent review of an important industrial problem and it should be published as it is, after (perhaps) some Editorial control of the English.
Author Response
This paper is a review of a seemingly subordinate, but in fact very dangerous aspect of cosmetics industry: the addition of hormones to cosmetics products. These additives could cause some short-time apparent advantages in the applicative goals of these products, but their long-term health effects are (could be) sources of serious health risks, including skin cancer and similar problems. The hormone additives in cosmetics represent a difficult analytical challenge: it is the detection/determination of small quantities in a fairly complicated mixture of (mainly organic) components. The present review gives a clear description of the background and of the major analytical techniques which can be used for the detection and/or determination of these additive, which often became applied illegally. The analysis requires two steps: sample preparation (separation) and the following determination of the dangerous substance. In the present review also the most modern techniques are described, as ultrasound-assisted extraction, solid-phase extraction or microextraction, together with frontier determination techniques, as immunoassay. The review is based on 100 well selected recent references, all are after 2000, mostly from the last decade. Summarizing the opinion of the present Referee: this is an excellent review of an important industrial problem and it should be published as it is, after (perhaps) some Editorial control of the English.
Response: Thank you so much for your positive comments. We have tried our best to revise our manuscript carefully and hope that the correction will meet with your approval.
Reviewer 2 Report
This article is well written (Advances on Hormones in Cosmetics: Illegal Addition Status, 2 Sample Preparation, and Detection Technology), however here are a few suggestions that will improve the quality of the manuscript if followed by the authors
The abstract is fine but it needs to be a bit more focused on the aim sentences, the conclusion part of the abstract is missing.
Add conclusive line in the end of abstract that should summarize your whole work.
Keep the introduction with recent supportive findings, if you can find some relevant intro of the main title recently published in 2022, that could be much better.
Please add a paragraph about illegal cosmetics
What is the aim of the manuscript? What news will it present? Will it attempt to answer any outstanding questions? If so, which ones? Can you write a paragraph in the introduction to grab the reader’s attention?
Please mention the full form first and then abbreviations in the following text of the manuscript
No need to add figure 1 in this way it should be revised.
Please provide references in the table 1.
Cosmetic sample preparation technologies must be elaborative along with their proper methods.
Rewrite conclusions and specifically focus on the main findings and give your recommendations in it.
Grammatically it needs serious attention.
Recheck your references according to the journal format.
Author Response
This article is well written (Advances on Hormones in Cosmetics: Illegal Addition Status, 2 Sample Preparation, and Detection Technology), however here are a few suggestions that will improve the quality of the manuscript if followed by the authors.
Response: Thank you so much for your positive comments. We have tried our best to revise our manuscript carefully according to your suggestion.
Question 1. The abstract is fine but it needs to be a bit more focused on the aim sentences, the conclusion part of the abstract is missing. Add conclusive line in the end of abstract that should summarize your whole work.
Response: Thank you very much for your advice. We have rewritten the abstract based on the comments. The changes have been marked up using the “Track Changes” in the revised manuscript.
Question 2. Keep the introduction with recent supportive findings, if you can find some relevant intro of the main title recently published in 2022, that could be much better.
Response: Thank you very much for your advice. We have added the literature published in 2022 in the revised manuscript.
Question 3. Please add a paragraph about illegal cosmetics.
Response: Thank you very much for your advice. In the Introduction, we have added the corresponding parts to talk about illegal cosmetics.
Question 4. What is the aim of the manuscript? What news will it present? Will it attempt to answer any outstanding questions? If so, which ones? Can you write a paragraph in the introduction to grab the reader’s attention?
Response: Thank you very much for your advice. We have rewritten the purpose of this review in the last paragraph of the Introduction. The changes have been marked up using the “Track Changes” in the revised manuscript.
Question 5. Please mention the full form first and then abbreviations in the following text of the manuscript
Response: Thank you so much for your kind suggestion. We have checked the abbreviations carefully based on the comments.
Question 6. No need to add figure 1 in this way it should be revised.
Response: Thank you for your advice. We have moved Figure 1 to the end of the Introduction. If you still think it is not suitable, we can adjust it to the graphic abstract.
Question 7. Please provide references in the table 1.
Response: Thank you for your suggestion, we have added references in Table 1.
Question 8. Cosmetic sample preparation technologies must be elaborative along with their proper methods.
Response: Thank you for your suggestion, we have added the corresponding methods in the revised manuscript.
Question 9. Rewrite conclusions and specifically focus on the main findings and give your recommendations in it.
Response: Thank you for your suggestion. We have rewritten the section of Conclusions and Outlooks based on the comments.
Question 10. Grammatically it needs serious attention.
Response: Thank you so much for your advice. We have checked the grammar carefully and polished this manuscript based on the comments.
Question 11. Recheck your references according to the journal format.
Response: Thank you very much for your advice. In the revised manuscript, we have rechecked the references according to the journal format.
Reviewer 3 Report
Effective surveillance and detection of hormones in cosmetics is of paramount importance because cosmetic safety is a key issue for consumers. The paper presents data on the addition of hormones in unqualified cosmetics in China from 2017 to 2022, detection values ​​and types of cosmetics, as well as hormonal threats in cosmetics for humans were discussed. The paper presents advanced technologies for the preparation of hormone samples in cosmetics in the years 2012-2022, as well as their new trends in the future. The work constituted interesting cognitive material that will interest readers. However, the work requires corrections before publication:
· Line 11 replace hormone with hormones
· Lines 16-17 replace ultrasound assisted extraction with ultrasound-assisted extraction
· Line 25-26 „With the rapid economic development and consumer upgrading, cosmetics have gradually become the daily necessity in people's lives.” Please consider making the sentence look like this: With the rapid economic development, greater consumer awareness, and the trend of taking care of health and body, the community is starting to use more and more cosmetics.
· Figure 1 should be placed after the part of the text in which it is mentioned. Abbreviations of techniques and methods for sample preparation and analyte determination are listed in Figure 1. Authors should use these abbreviations in the main text when describing the techniques and methods in section 1. Introduction
· Line 118 replace hormones used with hormones are used
· Line 120 replace cosmetics market with the cosmetics market
· Lines 131-132 replace Chinese National Medical Products Administration with the Chinese National Medical Products Administration
· Line 145 replace monitor with monitoring
· Line 158 replace ultrasound assisted extraction with ultrasound-assisted extraction
· Line 192 replace mainly with to main
· Line 192 replace mostly with most
· Line 211 replace cost long time and large with cost a long time and a large
· Line 225 repalce showed higher adsorption with showed a higher adsorption
· Line 236 replace has low with has a low
· Line 255 replace as extraction medium with as an extraction medium
· Line 276 replace firstly with first
· Line 279 replace dissolving or dispersing samples with dissolving or dispersing of samples
· Line 309 replace table with tables
· Line 318 replace manily with main
· Line 325 replace method which molecular imprinted monolithic column with method in which molecular imprinted monolithic column
· Line 326 replace were with was
· Line 418 replace new challenge with new challenges

Author Response
Effective surveillance and detection of hormones in cosmetics is of paramount importance because cosmetic safety is a key issue for consumers. The paper presents data on the addition of hormones in unqualified cosmetics in China from 2017 to 2022, detection values and types of cosmetics, as well as hormonal threats in cosmetics for humans were discussed. The paper presents advanced technologies for the preparation of hormone samples in cosmetics in the years 2012-2022, as well as their new trends in the future. The work constituted interesting cognitive material that will interest readers. However, the work requires corrections before publication.
Response: Thank you so much for your comments. We have tried our best to revise our manuscript carefully according to your suggestion.
Question 1. Line 11 replace hormone with hormones.
Response: Thanks for your comments, we have replaced hormone with hormones.
Question 2. Lines 16-17 replace ultrasound assisted extraction with ultrasound-assisted extraction
Response: Thanks for your comments, we have made changes based on the comments.
Question 3. Line 25-26 With the rapid economic development and consumer upgrading, cosmetics have gradually become the daily necessity in people's lives.” Please consider making the sentence look like this: With the rapid economic development, greater consumer awareness, and the trend of taking care of health and body, the community is starting to use more and more cosmetics.
Response: Thank you for your valuable comments, we have made changes to this sentence based on the comments.
Question 4. Figure 1 should be placed after the part of the text in which it is mentioned. Abbreviations of techniques and methods for sample preparation and analyte determination are listed in Figure 1. Authors should use these abbreviations in the main text when describing the techniques and methods in section 1. Introduction
Response: Thank you very much for your suggestion. In the revised manuscript, we have placed Figure 1 after the part of the text, and we have added the abbreviations in the section of the Introduction.
Question 5. Line 118 replace hormones used with hormones are used
Response: We agree with your comments and have made changes based on the comments.
Question 6. Line 120 replace cosmetics market with the cosmetics market
Response: Thanks for your comments, we have replaced cosmetics market with the cosmetics market.
Question 7. Lines 131-132 replace Chinese National Medical Products Administration with the Chinese National Medical Products Administration
Response: We agree with your comments and have made changes based on the comments.
Question 8. Line 145 replace monitor with monitoring
Response: Thanks for your comments, we have replaced monitor with monitoring.
Question 9. Line 158 replace ultrasound assisted extraction with ultrasound-assisted extraction
Response: We agree with your comments and have made changes based on the comments.
Question 10. Line 192 replace mainly with to main
Response: Thanks for your suggestion, we have made changes based on the comments.
Question 11. Line 192 replace mostly with most
Response: Thanks for your comments, we have replaced mostly with most.
Question 12. Line 211 replace cost long time and large with cost a long time and a large
Response: We agree with your comments and have made changes based on the comments.
Question 13. Line 225 replace showed higher adsorption with showed a higher adsorption
Response: Thanks for your suggestion, we have made changes based on the comments.
Question 14. Line 236 replace has low with has a low
Response: Thanks for your suggestion, we have made changes based on the comments.
Question 15. Line 255 replace as extraction medium with as an extraction medium
Response: We have replaced as extraction medium with as an extraction medium.
Question 16. Line 276 replace firstly with first
Response: We agree with your comments and have made changes based on the comments.
Question 17. Line 279 replace dissolving or dispersing samples with dissolving or dispersing of samples
Response: Thanks for your comments, we have replaced dissolving or dispersing samples with dissolving or dispersing of samples.
Question 18. Line 309 replace table with tables
Response: We agree with your comments and have made changes based on the comments.
Question 19. Line 318 replace mainly with main
Response: Thanks for your comments, we have replaced mainly with main.
Question 20. Line 325 replace method which molecular imprinted monolithic column with method in which molecular imprinted monolithic column
Response: We agree with your comments and have made changes based on the comments.
Question 21. Line 326 replace were with was
Response: Thanks for your comments, we have replaced were with was.
Question 22. Line 418 replace new challenge with new challenges
Response: We agree with your comments and have made changes based on the comments.
Round 2
Reviewer 2 Report
Article has been well revised,
Can you please provide the separate files of all figures with good results.